# Using smartphone step counts to monitor patients with total hip arthroplasty: The impact of patients' living arrangements and residential location

Satoshi Yamate[1,2], Satoshi Hamai[1*], Toshiki Konishi[1], Yuki Nakao[1], Takahiro Inoue[1], Goro Motomura[1], Yasuharu Nakashima[1]

1 Department of Orthopedic Surgery, Graduate School of Medical Sciences, Kyushu University, Fukuoka, Japan, 2 Department of Orthopaedic Surgery and Rheumatology, National Hospital Organization Kyushu Medical Center, Fukuoka, Japan

* hamai.satoshi.075@m.kyushu-u.ac.jp

## Abstract

### Background

Smartphone step counts may capture real-world activities in patients' daily lives, and the self-monitoring of step counts may introduce healthier behavioral changes. We investigated the association between living arrangements (solitude/cohabiting) and residential location (urban/suburban) of patients and their preoperative and postoperative smartphone step counts following total hip arthroplasty (THA).

### Materials and methods

Patients scheduled for THA at a university hospital from September 10, 2021, to December 5, 2023 were enrolled into the study from August 4, 2021, to November 10, 2023. We remotely monitored the patients' daily step counts from smartphone application until up to 365 days after THA. We used the moving average method and latent growth curve modeling to analyze the time-series data of the step count.

### Results

Overall, 85 patients were included in this study. Comparing the 37 solitary and 48 cohabiting patients, the percentage of men was higher in the solitary group (27% vs. 8%, $P=0.037$). There were no notable differences in the demographics between the 44 urban and 41 suburban patients. Urban patients experienced a 483-step per 2-week greater preoperative increase than suburban patients ($P=0.027$) after adjusting for confounding factors. From postoperative 2 weeks to preoperative 12 weeks, the intercept was larger for urban patients than for suburban patients by 893 steps ($P=0.040$). From

**Data availability statement:** Data cannot be shared publicly because of the need for consent from study participants. Data are available from the Kyushu University Hospital Institutional Review Board (ijkseimei@jimu.kyushu-u.ac.jp) for researchers who meet the criteria for access to confidential data.

**Funding:** This work was supported by the Japan Society for the Promotion of Science (KAKENHI) (https://www.jsps.go.jp/, grant number JP23K08654) awarded to YN, and Medical Care Education Research Foundation (http://mcef.or.jp/index.html) awarded to SH. The funders had no role in study design, data collection and analysis, decision to publish, or preparation of the manuscript.

**Competing interests:** I have read the journal's policy and the authors of this manuscript have the following competing interests: Satoshi Hamai received speakers bureaus from Stryker, Johnson & Johnson, Japan MDM, KAKEN PHARMACEUTICAL CO., LTD. Yasuharu Nakashima received research grants from Chugai Pharmaceutical Co., Ltd., AYUMI Pharmaceutical Corporation, EA Pharma Co., Ltd., KYOCERA Corporation, and Zimmer Biomet G.K. All other authors have no conflicts of interest to disclose. This does not alter our adherence to PLOS ONE policies on sharing data and materials.

postoperative 20 weeks to preoperative 50 weeks, the intercept was larger for solitary patients than for cohabiting patients by a difference of 1,360 steps ($P = 0.027$).

## Conclusions

Living arrangements and residential locations were associated with daily step counts before and after THA. Patients in urban areas had higher step counts after the initiation of step-count monitoring and during the early postoperative period. Solitary patients walked more than cohabiting patients. Our findings underscore the utility of smartphone step counts as objective outcome measures for patient assessment and encouraging healthier behavioral changes.

---

## Introduction

Smartphone-based care platforms for total hip arthroplasty (THA) can play a critical role in modern orthopaedic care [1]. Remote monitoring using mobile health technology has enabled capturing real-world activities of daily living [2]. This technology can also facilitate behaviour change techniques through prompt self-monitoring of behavior rather than mere measurement [3]. Self-monitoring is a simple and inexpensive technique and can drive patients towards healthier behaviours when monitoring is collaboratively managed with healthcare professionals [4,5].

Step count is the most fundamental indicator beyond a physical activity measure and can quantitatively encompass physical health. Higher daily step counts are reportedly associated with a reduced risk of chronic diseases [6] and lower all-cause mortality [7]. Many patients own smartphones, which might be useful for capturing real-world step counts in the social settings in which they live and could be used to reveal health disparities by focusing on activity levels during the perioperative THA period.

Concerns of solitude among the older population worldwide are increasing [8]. However, the impact of patients' living arrangements on perioperative activity in THA is unknown. Urban environments significantly affect physical activity [9]. Previous studies have consistently shown that large cities have easily walkable areas, and that urban citizens have a positive attitude towards walking and prioritize a pleasant walking environment [10]. Therefore, living arrangements and the residential location could also be a crucial factor affecting perioperative activity in THA; however, to the best of our knowledge, these associations have not been previously reported.

This study investigated the association between living arrangements, residential locations, and preoperative and postoperative daily step counts in patients who underwent THA. We hypothesized that living arrangements and residential locations are associated with daily step count from a THA patient's smartphone.

## Materials and methods

### Study design and setting

This observational cohort study was approved by the Kyushu University Hospital Institutional Review Board for Clinical Research (Registration number: 2021–197). It

was conducted in accordance with the Strengthening the Reporting of Observational Studies in Epidemiology (STROBE) guidelines [11] and ethical standards of the Declaration of Helsinki. Written informed consent was obtained from all participants prior to their inclusion in the study.

## Participants

We identified patients scheduled for THA from September 10, 2021, to December 5, 2023 and obtained their consent for participation at the time of preoperative orientation, approximately 1–3 months before surgery. Accordingly, the recruitment period—defined as the time during which participants were formally enrolled in the study—was from August 4, 2021 to November 10, 2023. After obtaining a completed written consent form, we installed the application (mymobility, Zimmer Biomet) [1,5,12] on patient-owned smartphones with the assistance of a medical assistant. We wished to analyse real-world data obtained in natural circumstances, not in a controlled clinical trial. Therefore, we let the patients decide how they wanted to use the application. Step count monitoring was conducted by an author who was not an attending physician to prevent intervention bias, and no direct feedback was provided to the patient. THA was performed using a posterolateral approach with a cementless stem and acetabular component in all patients by the same group of nine senior surgeons. In principle, the clinical path was adapted to discharge patients home 2 weeks after surgery. Those who wished were transferred to a rehabilitation hospital and continued inpatient rehabilitation for maximum 3 months before discharge to home.

## Variables of grouping

We focused on the living arrangement and residential location of patients based on information from hospital admission surveys. For living arrangements, a binary variable (solitude/cohabiting) was used to indicate whether the patient lived alone or cohabited. For residential location, we used a binary variable (urban/suburban) indicating whether the patient lived in an ordinance-designated city with over 500,000 residents, as recognized by the Japanese government for increased autonomy [13], or other suburban areas.

## Outcome measures

**Daily step count.** Smartphones are reportedly accurate in tracking step counts and only slightly differ from the observed step counts [14]. Daily step counts were sequentially obtained remotely from smartphone app registration before surgery until January 11, 2024, or up to 365 days postoperatively.

Daily step count reportedly demonstrates significant inter- and intraindividual variability over time [15]. Thus, a mean of ≥3 days is recommended when tracking daily steps via smartphones to measure the daily step count [16]. We used the mean daily step count data from the 7-day window for each time point, including 3 days before and 3 days after, to measure the daily step count.

**Other outcomes.** For additional analysis, we collected the Oxford Hip Score (OHS) and Hip Disability and Osteoarthritis Outcome Score (HOOS), which are patient-reported outcome measures (PROMs) for hip osteoarthritis, using the validated Japanese versions [17]. The OHS ranges from 0 to 48, with higher scores indicating better outcomes. HOOS is based on five subscales for symptoms, pain, activities of daily living (ADL), sports, and quality of life (QOL); the score ranges from 0 to 100, with higher scores indicating better outcomes. The reliability and validity of OHS and HOOS have been established for Japanese patients undergoing THA. We obtained OHS and HOOS via smartphone preoperatively and at 1 month, 3 months, 6 months, and 1 year postoperatively.

We also investigated the number of postoperative days before the physical therapist allowed cane-walking independence, length of hospital stay, and transfer rate to the rehabilitation hospital.

**Confounding factors.** Confounding factors included patient age, sex, body mass index (BMI), American Society of Anesthesiologists physical status (ASA-PS) [18], and smartwatch use in conjunction with a smartphone, which could affect step count measurement. These confounding variables were obtained during preoperative THA assessment.

## Sample size

We allowed an error margin of up to 1,000 steps for step count estimation. Considering the standard deviation for Japanese step count is approximately 3,000 steps [19,20], the sample size needed to fit within the acceptable range of 95% confidence intervals was ≥ 35 patients per group.

## Statistical analyses

Fisher's exact test was used for categorical variables, and Welch's *t*-test was used for continuous variables among multiple groups. The analysis was divided into three time periods before and after THA: Model 1 (8, 6, 4, and 2 weeks preoperatively), Model 2 (2, 4, 6, 8, 10, and 12 weeks postoperatively), and Model 3 (20, 30, 40, and 50 weeks postoperatively). The daily step number at each time point was described and compared between the groups.

We used latent growth curve modelling to interpret changes over time in perioperative step counts in THA patients. Latent growth curve modelling is a time-series analysis method that captures temporal changes in repeated measurement data and has been widely used for an extended period in developmental psychology, education, and social sciences [21]. This method assumes hypothetical variables (latent variables), such as a common intercept and slope for repeated measurement outcomes, separates the collective trends and individual differences in temporal changes, and estimates the coefficients for the latent variables (Fig 1).

All analyses were conducted using R version 4.3.1 (R Foundation for Statistical Computing, Vienna, Austria) and Python version 3.12.0 (Python Software Foundation, Wilmington, DE, USA). Latent growth curve modelling is available in the free and open-source package lavaan. Since there was no evidence that the missing values were missing completely at random, multiple imputations were performed to include patients with missing values using the mice package [22]. We used the Centered moving average [23], a method used in the data smoothing process, particularly for time series data.

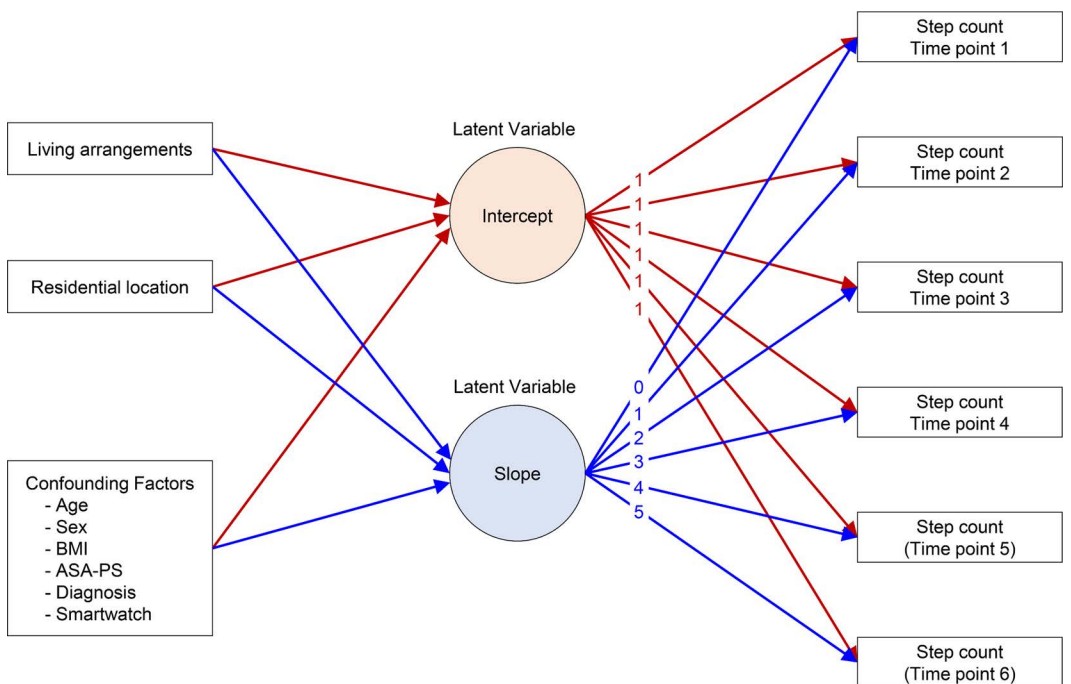

**Fig 1. Path diagram assumed in this study illustrating the relationship among the variables of interest, confounders, latent variables, and step count for latent growth curve modelling. BMI, body mass index; ASA-PS, American Society of Anesthesiologists physical status.**

The time trends regarding daily step counts were drawn as Centered 7-day moving averages with 95% confidence intervals using the Pandas and Matplotlib packages. Statistical significance was set at *P*<0.05.

## Results

Overall, 233 patients were assessed for eligibility. Among them, 60 did not have smartphones or were unable to install the application, leaving 173 patients (74.2%) with compatible smartphones. Of these, 54 declined to participate, and 119 wished to participate in the study. Among the 119 patients, data from 34 were excluded due to incomplete or invalid step-count data. Finally, data from 85 patients who underwent THA and had valid step-count data were included in the analysis (Fig 2). The mean age of the 85 patients was 61.7 years, significantly younger than the 66.4 years of the 148 excluded patients (*P*<0.001; S1 Table).

Of the 85 patients, 37 were living in solitude, and 48 were cohabiting, with a higher percentage of men among those in solitude (27% vs. 8%, *P*=0.037). Regarding residential location, 44 patients lived in urban areas, and 41 lived in suburban areas, with no notable differences in demographics (Table 1).

The daily step count of the 85 patients increased after preoperative registration to 3,082 steps 2 weeks before surgery (Table 2; S2 Table). The number of steps decreased immediately after THA but recovered to 3,332 steps 6 weeks after surgery, exceeding the preoperative level, and continued to increase (S1 Fig). The number of steps taken over time differed between groups according to living arrangements (Fig 3) and residential locations (Fig 4). At the same time points, solitary patients had more steps than cohabiting patients generally for the entire observation period, with significant differences at preoperative 2 weeks (*P*=0.010), postoperative 4 weeks (*P*=0.039), 8 weeks (*P*=0.010), 10 weeks (*P*=0.026),

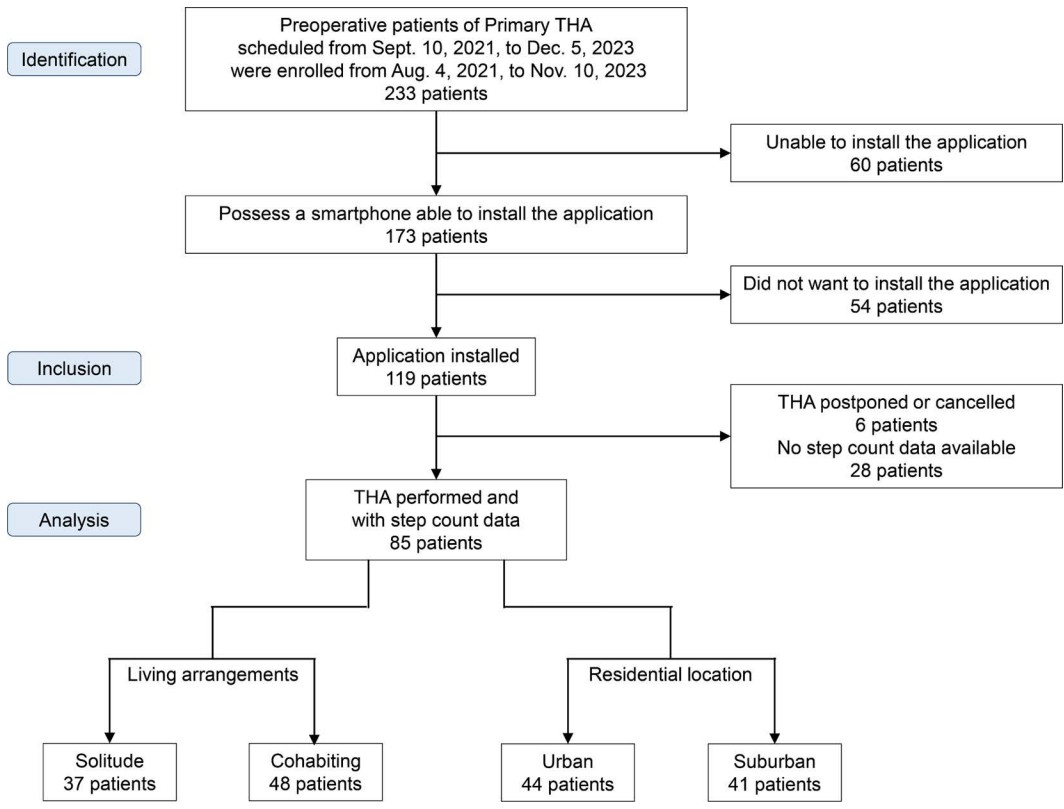

**Fig 2. Flow diagram.** THA, total hip arthroplasty.

**Table 1. Patient demographics grouped by living arrangements.**

| Variables | Overall (n = 85) | Solitude (n = 37) | Cohabiting (n = 48) | P value | Urban (n = 44) | Suburban (n = 41) | P value |
|---|---|---|---|---|---|---|---|
| Age at survey (*yr* [SD]) | 61.7 (9.4) | 60.4 (8.2) | 62.7 (10.1) | 0.257 | 60.4 (10.2) | 63.0 (8.3) | 0.207 |
| Sex (*no.* [%]) | | | | 0.037† | | | 0.773 |
| Men | 14 (17) | 10 (27) | 4 (8) | | 8 (18) | 6 (15) | |
| Women | 71 (84) | 27 (73) | 44 (92) | | 36 (82) | 35 (85) | |
| BMI (*kg/m²* [SD]) | 24.9 (4.3) | 23.8 (4.0) | 25.7 (4.4) | 0.042† | 25.2 (4.5) | 24.5 (4.2) | 0.422 |
| Diagnosis (*no.* [%]) | | | | 0.706 | | | 0.417 |
| OA | 73 (86) | 30 (81) | 43 (90) | | 38 (86) | 35 (85) | |
| ONFH | 5 (6) | 3 (8) | 2 (4) | | 4 (9) | 1 (2) | |
| RA | 4 (5) | 2 (5) | 2 (4) | | 1 (2) | 3 (7) | |
| SIF | 3 (4) | 2 (5) | 1 (2) | | 1 (2) | 2 (5) | |
| Residential location (*no.* [%]) | | | | 0.275 | | | <0.001† |
| Urban | 44 (52) | 22 (59) | 22 (46) | | 44 (100) | 0 (0) | |
| Suburban | 41 (48) | 15 (41) | 26 (54) | | 0 (0) | 41 (0) | |
| Living arrangements (*no.* [%]) | | | | <0.001† | | | 0.275 |
| Solitude | 37 (44) | 37 (100) | 0 (0) | | 22 (50) | 15 (37) | |
| Cohabiting | 48 (56) | 0 (0) | 48 (100) | | 22 (50) | 26 (63) | |
| Preoperative Oxford Hip Score (SD) | 26.4 (8.7) | 28.5 (8.7) | 25.3 (8.7) | 0.266 | 26.1 (8.3) | 26.6 (9.1) | 0.855 |

SD, standard deviation; BMI, body mass index; OA, osteoarthritis; ONFH, osteonecrosis of the femoral head; RA, rheumatoid arthritis; SIF, subchondral insufficiency fracture

†Significant at P < 0.05

**Table 2. Comparison of mean daily step count at each time point.**

| Variables | Overall (n = 85) | Solitude (n = 37) | Cohabiting (n = 48) | P value | Urban (n = 44) | Suburban (n = 41) | P value |
|---|---|---|---|---|---|---|---|
| **Period of Model 1** | | | | | | | |
| Preoperative 8 weeks | 2,133 (1,988) | 2,699 (2,607) | 1,621 (1,8) | 0.104 | 2,127 (1,806) | 2,137 (2,168) | 0.987 |
| Preoperative 6 weeks | 2,446 (2,213) | 2,972 (2,186) | 1,997 (2,175) | 0.122 | 2,859 (2,486) | 2,033 (1,861) | 0.190 |
| Preoperative 4 weeks | 2,866 (2,494) | 3,242 (2,539) | 2,558 (2,452) | 0.296 | 3,365 (2,894) | 2,256 (1,762) | 0.073 |
| Preoperative 2 weeks | 3,082 (2,192) | 3,960 (2,174) | 2,525 (2,038) | 0.010† | 3,910 (2,366) | 2,279 (1,683) | 0.002† |
| **Period of Model 2** | | | | | | | |
| Postoperative 2 weeks | 2,655 (2,838) | 3,336 (3,111) | 2,139 (2,534) | 0.103 | 3,280 (3,627) | 2,119 (1,815) | 0.120 |
| Postoperative 4 weeks | 2,799 (2,430) | 3,614 (2,943) | 2,249 (1,858) | 0.039† | 3,251 (2,820) | 2,433 (2,029) | 0.188 |
| Postoperative 6 weeks | 3,332 (2,490) | 4,177 (2,989) | 2,798 (1,979) | 0.053 | 4,127 (2,810) | 2,718 (2,049) | 0.033† |
| Postoperative 8 weeks | 3,664 (2,423) | 4,677 (2,575) | 2,988 (2,089) | 0.010† | 4,191 (2,685) | 3,261 (2,155) | 0.155 |
| Postoperative 10 weeks | 3,697 (2,550) | 4,712 (2,935) | 3,041 (2,054) | 0.026† | 4,318 (2,877) | 3,197 (2,171) | 0.114 |
| Postoperative 12 weeks | 3,801 (2,678) | 4,739 (3,034) | 3,175 (2,247) | 0.046† | 4,197 (2,723) | 3,471 (2,640) | 0.323 |
| **Period of Model 3** | | | | | | | |
| Postoperative 20 weeks | 4,316 (2,914) | 5,345 (3,102) | 3,575 (2,585) | 0.057 | 4,513 (2,931) | 4,089 (2,953) | 0.640 |
| Postoperative 30 weeks | 4,425 (3,924) | 6,660 (4,791) | 2,967 (2,380) | 0.012† | 4,068 (3,084) | 4,746 '4,610) | 0.595 |
| Postoperative 40 weeks | 4,477 (2,874) | 5,639 (3,676) | 3,606 (1,719) | 0.062 | 4,785 (2,379) | 4,187 (3,318) | 0.543 |
| Postoperative 50 weeks | 4,006 (2,939) | 5,397 (3,515) | 2,893 (1,796) | 0.017† | 4,463 (2,816) | 3,549 (3,068) | 0.358 |

Mean value and standard deviation in parentheses.

†Significant at *P* < 0.05.

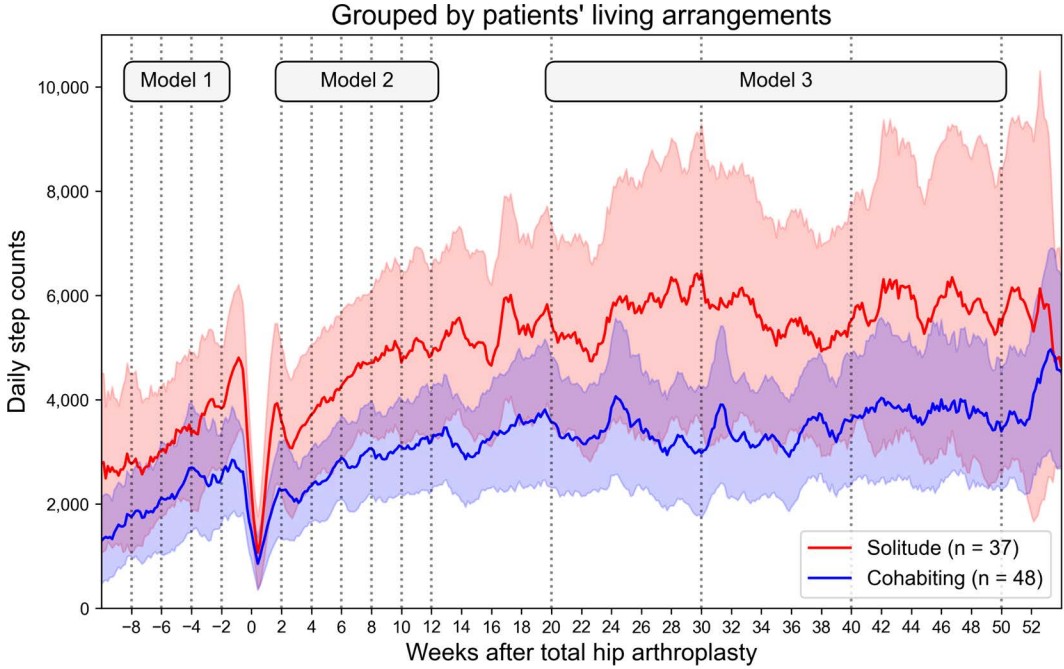

**Fig 3. Cantered 7-day moving average of daily step counts grouped by patients' living arrangements.** The bands indicate 95% confidence intervals.

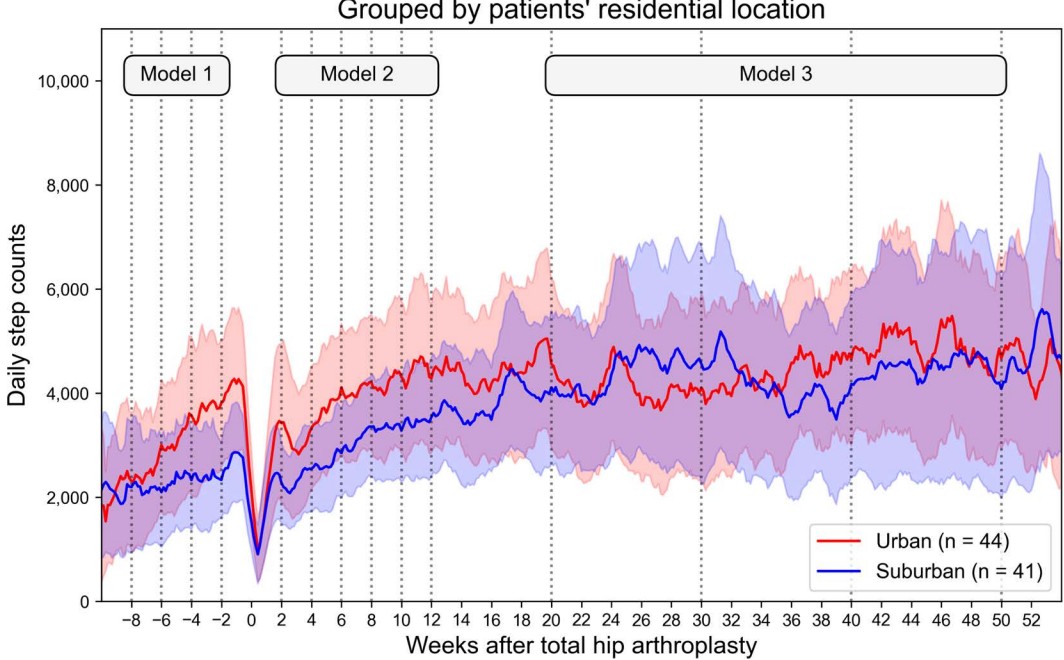

**Fig 4. Cantered 7-day moving average of daily step counts grouped by patients' residential location.** The bands indicate 95% confidence intervals.

12 weeks ($P=0.046$), 30 weeks ($P=0.012$), and 50 weeks ($P=0.017$). Regarding residential location, urban patients walked more than suburban patients for preoperative 2 weeks ($P=0.002$) and postoperative 6 weeks ($P=0.033$).

Longitudinal analyses adjusted for confounding factors using the latent growth curve model (Model 1) showed that the increase in preoperative step count was higher in urban patients than in suburban patients, with an estimated difference of 483 steps per 2 weeks ($P=0.027$, Table 3). From postoperative 2 weeks to preoperative 12 weeks (Model 2), the intercept was larger for urban patients than for suburban patients, with an estimated difference of 893 steps ($P=0.040$; Table 4), with no difference in the slope. From postoperative 20 weeks to preoperative 50 weeks (Model 3), the intercept was larger

**Table 3. Regression coefficients for latent variables of step count from preoperative 8 weeks to preoperative 2 weeks (Model 1).**

| Variables | β (95% CI) for intercept | P value | β (95% CI) for slope | P value |
|---|---|---|---|---|
| Living arrangements | | | | |
| Solitude (Reference: Cohabiting) | 70 (−788, 929) | 0.872 | 93 (−347, 533) | 0.678 |
| Residential location | | | | |
| Urban (Reference: Suburban) | 84 (−751, 920) | 0.843 | 483 (55, 911) | 0.027[†] |
| Confounding factors | | | | |
| Age at survey | −28 (−76, 20) | 0.252 | −2 (−26, 23) | 0.900 |
| Sex | | | | |
| Women (Reference: Men) | −700 (−1929, 530) | 0.264 | 137 (−494, 768) | 0.670 |
| BMI ($kg/m^2$) | −81 (−192, 30) | 0.154 | 4 (−53, 61) | 0.891 |
| Diagnosis | | | | |
| OA (Reference: Others) | −175 (−1489, 1139) | 0.793 | −90 (−763, 582) | 0.792 |
| ASA-PS | −521 (−1441, 398) | 0.266 | −253 (−724, 219) | 0.293 |
| Smart watch user | −882 (−1874, 111) | 0.082 | 456 (−53, 965) | 0.079 |

CI, confidence interval; BMI, body mass index; OA, osteoarthritis; Others, osteonecrosis of the femoral head, rheumatoid arthritis, and subchondral insufficiency fracture; ASA-PS, American Society of Anesthesiologists physical status

[†]Significant at $P<0.05$.

**Table 4. Regression coefficients for latent variables of step count from postoperative 2 weeks to postoperative 12 weeks (Model 2).**

| Variables | β (95% CI) for intercept | P value | β (95% CI) for slope | P value |
|---|---|---|---|---|
| Living arrangements | | | | |
| Solitude (Reference: Cohabiting) | 530 (−346, 1407) | 0.235 | 90 (−270, 451) | 0.624 |
| Residential location | | | | |
| Urban (Reference: Suburban) | 893 (40, 1745) | 0.040[†] | −6 (−356, 345) | 0.974 |
| Confounding factors | | | | |
| Age at survey | −18 (−67, 32) | 0.480 | −4 (−25, 16) | 0.681 |
| Sex | | | | |
| Women (Reference: Men) | −673 (−1926, 581) | 0.293 | 133 (−383, 649) | 0.613 |
| BMI ($kg/m^2$) | 2 (−111, 116) | 0.968 | 2 (−45, 49) | 0.927 |
| Diagnosis | | | | |
| OA (Reference: Others) | −338 (−1675, 1000) | 0.620 | −9 (−559, 541) | 0.974 |
| ASA-PS | −1507 (−2447, −566) | 0.002[†] | −32 (−419, 355) | 0.871 |
| Smart watch user | 1558 (545, 2571) | 0.003[†] | −294 (−711, 123) | 0.167 |

Latent growth curve modeling was used for the analysis. CI, confidence interval; BMI, body mass index; OA, osteoarthritis; Others, osteonecrosis of the femoral head, rheumatoid arthritis, and subchondral insufficiency fracture; ASA-PS, American Society of Anesthesiologists physical status

[†]Significant at $P<0.05$.

for solitary patients than for cohabiting patients, with an estimated difference of 1,360 steps (*P*=0.027, Table 5), while there was no difference in slope.

The two groups of solitude and cohabiting patients had no differences in the preoperative OHS and HOOS. Solitude and cohabiting patients showed a similar trend postoperatively, and the difference was insignificant (S2 Fig). The number of suburban patients was higher than that of urban patients postoperatively and at a significant level for HOOS-QOL at 1 month (51.9 vs 64.2, *P*=0.025) and 3 months (73.4 vs. 59.2, *P*=0.010), HOOS-pain at 1 year (83.6 vs 68.2, *P*=0.047, S3 Fig), and OHS at 1 year (44.8 vs. 39.8, *P*=0.027, S4 Fig). The two groups of urban and suburban patients had no significant differences in postoperative days required to cane walking independence, length of stay, or transfer rate to a rehabilitation hospital (S3 Table).

## Discussion

We found that living arrangements and residential locations were associated with variations in step count and had different effects depending on the THA perioperative period. Solitude patients maintained higher activity levels from the preoperative to the postoperative period than cohabiting patients did, a difference that was not captured by OHS or HOOS. Urban patients walked more than suburban patients preoperatively after registering for the application and took more steps in the early postoperative period.

Our study revealed that living alone was not a negative factor for activity level. Fleischman et al reported no increase in post-discharge complications or unplanned clinical events in patients living alone [24], which is consistent with our findings. Japan is gradually experiencing a breakdown in its traditional social structure and increasing social isolation in the older population and is said to be the most socially isolated population in the world today [25]. Our study found that solitary patients exhibited significantly higher step counts than cohabiting patients during the plateau phase of postoperative recovery, which spanned from postoperative week 20 to week 50, suggesting that living alone may require greater physical autonomy in daily life. This finding may provide helpful information for surgical decision-making. According to the 2021 Japanese government survey [26] of citizens aged 65 and over, solitude among women was 22.8%, and solitude among men was 15.3%, indicating a higher rate of solo living among women. However, we found that solitary patients included more men than cohabiting patients. The reversal of the sex difference in THA patients living alone versus those cohabiting

**Table 5. Regression coefficients for latent variables of step count from postoperative 20 weeks to postoperative 50 weeks (Model 3).**

| Variables | β (95% CI) for intercept | *P* value | β (95% CI) for slope | *P* value |
|---|---|---|---|---|
| Living arrangements | | | | |
| Solitude (Reference: Cohabiting) | 1360 (156, 2565) | 0.027† | 46 (−577, 669) | 0.885 |
| Residential location | | | | |
| Urban (Reference: Suburban) | −176 (−1346, 995) | 0.768 | 16 (−589, 622) | 0.958 |
| Confounding factors | | | | |
| Age at survey | −27 (−95, 41) | 0.432 | −6 (−41, 29) | 0.746 |
| Sex | | | | |
| Women (Reference: Men) | 120 (−1602, 1843) | 0.891 | 122 (−769, 1013) | 0.788 |
| BMI (*kg/m²*) | −112 (−268, 44) | 0.160 | −3 (−84, 77) | 0.934 |
| Diagnosis | | | | |
| OA (Reference: Others) | 141 (−1696, 1979) | 0.880 | −143 (−1092, 806) | 0.768 |
| ASA-PS | −1784 (−3077, −491) | 0.007† | 384 (−285, 1052) | 0.260 |
| Smart watch user | 174 (−1218, 1567) | 0.806 | −97 (−817, 624) | 0.792 |

Latent growth curve modeling was used for the analysis. CI, confidence interval; BMI, body mass index; OA, osteoarthritis; Others, osteonecrosis of the femoral head, rheumatoid arthritis, and subchondral insufficiency fracture; ASA-PS, American Society of Anesthesiologists physical status

†Significant at *P*<0.05.

with the general population is interesting, and there are several possible hypotheses. For example, living alone versus cohabiting is associated with disease through lifestyle or access to healthcare, which requires future studies.

A previous systematic review reported that self-monitoring step counts in patients with cardiovascular diseases increased by 2,503 steps/day [4]. Using time-series data, we found that the effect of prompt self-monitoring of behaviour via smartphone step counts was more remarkable in urban patients than in suburban patients. Urban populations are known to have higher health literacy than rural populations [27], which supports our findings. Perceived walkability is positively associated with the frequency of leisure-time physical activity in the general population [28]. Therefore, we considered urban environments an essential factor for health promotion, as previously reported [9]. A previous study reported that the length of hospital stay after THA was shorter for app users than for non-app users [5]. Our results suggest that this effect may depend on the patient's living environment. Self-monitoring and shared monitoring of step counts by patients and healthcare professionals may be effective for a broader population to introduce healthier behaviour changes.

Recently, PROMs have been emphasized for evaluation of THA outcome [17]. However, step count has not yet received attention as an outcome. In an adult US sample, more daily steps were significantly associated with lower all-cause mortality [7]. There was also a discrepancy between the subjective and objective activity levels after THA [29], which was also observed in our study. PROMs produce a ceiling effect in exchange for simplification [30]. In contrast, the step count is an objective and simple way to quantify the amount of activity, and no ceiling effect can occur. Combining step counts with PROMs could provide a more comprehensive and objective evaluation of THA outcomes in orthopaedic practice.

The study limitations include selection bias, as only those who installed the app were analysed. Another was misclassification bias. For example, remote areas within government-designated cities were not considered. Further, we accepted a wide error of 1000 steps in determining the sample size. Additionally, the step count via a smartphone requires the assumption that the user always carries the smartphone. Moreover, because we used a basic latent growth curve model, there may still be room for improvement in the fitting of the analytical model. Additionally, unmeasured confounding factors, such as comorbidities and the presence or absence of regular users, may have an impact. Despite these limitations, the strength is that this is the first study to visualize the association between patients' living environments and pre- and postoperative daily step counts in patients undergoing THA.

In conclusion, living arrangements and residential locations were associated with daily step counts before and after THA. Patients in urban areas had higher step counts after initiation of self-monitoring and during the early postoperative period. Solitude patients walked more than cohabiting patients. Our findings underscore the utility of smartphone step counts as objective outcome measures for patient assessment and encouraging healthier behavioral changes.

## Supporting information

**S1 Fig. Centered 7-day moving average of daily step counts across all patients.**
(DOCX)

**S2 Fig. Comparison of Oxford Hip Score by patients' living arrangements and residential location.**
(DOCX)

**S3 Fig. Comparison of Hip Disability and Osteoarthritis Outcome Score by patients' living arrangements.**
(DOCX)

**S4 Fig. Comparison of Hip Disability and Osteoarthritis Outcome Score by patients' residential location.**
(DOCX)

**S1 Table. Patient demographics of analyzed and excluded groups.**
(DOCX)

**S2 Table. Comparison of mean daily step count at each time point.**
(DOCX)

**S3 Table. Outcomes grouped by patients' living arrangements and residential location.**
(DOCX)

## Author contributions

**Conceptualization:** Satoshi Yamate, Toshiki Konishi, Yuki Nakao, Takahiro Inoue, Yasuharu Nakashima.

**Data curation:** Satoshi Yamate.

**Formal analysis:** Satoshi Yamate.

**Funding acquisition:** Satoshi Hamai, Yasuharu Nakashima.

**Investigation:** Satoshi Yamate, Toshiki Konishi, Yuki Nakao, Takahiro Inoue, Goro Motomura.

**Methodology:** Satoshi Yamate, Satoshi Hamai.

**Project administration:** Satoshi Yamate, Satoshi Hamai, Yasuharu Nakashima.

**Resources:** Satoshi Hamai.

**Software:** Satoshi Yamate.

**Supervision:** Satoshi Hamai, Yasuharu Nakashima.

**Validation:** Satoshi Yamate.

**Visualization:** Satoshi Yamate.

**Writing – original draft:** Satoshi Yamate.

**Writing – review & editing:** Satoshi Hamai, Toshiki Konishi, Yuki Nakao, Takahiro Inoue, Goro Motomura, Yasuharu Nakashima.

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
