## [Decision Letter · Decision Letter 0]

Dear Dr. Hamai,

Thank you for submitting your manuscript to PLOS ONE. After careful consideration, we feel that it has merit but does not fully meet PLOS ONE’s publication criteria as it currently stands. Therefore, we invite you to submit a revised version of the manuscript that addresses the points raised during the review process.

We look forward to receiving your revised manuscript.

Kind regards,

Osama Farouk

Academic Editor

PLOS ONE

Journal Requirements:

2**.** In this instance it seems there may be acceptable restrictions in place that prevent the public sharing of your minimal data. However, in line with our goal of ensuring long-term data availability to all interested researchers, PLOS’ Data Policy states that authors cannot be the sole named individuals responsible for ensuring data access (http://journals.plos.org/plosone/s/data-availability#loc-acceptable-data-sharing-methods).

Reviewers' comments:

Reviewer's Responses to Questions

**Comments to the Author**

1. Is the manuscript technically sound, and do the data support the conclusions?

Reviewer #1: Yes

Reviewer #2: Yes

2. Has the statistical analysis been performed appropriately and rigorously?

Reviewer #1: Yes

Reviewer #2: Yes

3. Have the authors made all data underlying the findings in their manuscript fully available?

Reviewer #1: Yes

Reviewer #2: Yes

4. Is the manuscript presented in an intelligible fashion and written in standard English?

Reviewer #1: Yes

Reviewer #2: Yes

Reviewer #1: The research is quite satisfactory as regards methodology and scientific writing. The discussion and conclusions are very useful especially the remark by the authors that there is a breakdown in its traditional social structure and increasing social isolation in the older population in their community and its potential effect on results after hip replacement surgery

Reviewer #2: Introduction

Are there any studies that investigated the relation between residence and step count after THA. If not, please, mention that in the introduction as this adds to the novelty of your paper

Patients and methods

Did you use a validated version of HOOS and Oxford for the Japanese population.

Results

Patient flowchart is not clear. How many patients were excluded and why?

Discussion

Could you elaborate more on why patients living in Solitude had more step count than cohabiting patients?

Was there any relation between PROMs and step count?

**Do you want your identity to be public for this peer review?** For information about this choice, including consent withdrawal, please see our Privacy Policy

Reviewer #1: No

Reviewer #2: No

---

## [Author Response · Author response to Decision Letter 1]

12 Apr 2025

Response to Reviewer #1:

Reviewer #1 (1): The research is quite satisfactory as regards methodology and scientific writing. The discussion and conclusions are very useful especially the remark by the authors that there is a breakdown in its traditional social structure and increasing social isolation in the older population in their community and its potential effect on results after hip replacement surgery

Response:

We thank the Reviewer #1 for the positive comments. We thank Reviewer #1 for the positive comments. This study is the first to highlight the potential influence of living arrangements and residential location on perioperative physical activity in patients undergoing THA. 

Response to Reviewer #2:

Reviewer #2 (1): Introduction

Are there any studies that investigated the relation between residence and step count after THA. If not, please, mention that in the introduction as this adds to the novelty of your paper

Response:

We are grateful to Reviewer #2 for critical comments and useful suggestions that helped us to improve our paper considerably. As suggested, we have revised the Introduction to clarify that this is the first study to investigate the association between residence and step count after THA.

Text Changes (Introduction, Lines 66-68):

Therefore, living arrangements and the residential location could also be a crucial factor affecting perioperative activity in THA; however, to the best of our knowledge, these associations have not been previously reported.

Reviewer #2 (2):

Patients and methods

Did you use a validated version of HOOS and Oxford for the Japanese population.

Response:

We thank Reviewer #2 for the valuable comment. Yes, we used validated Japanese versions of the HOOS and Oxford scores. We have revised the manuscript.

Text Changes (Materials and methods, Lines 123-125):

For additional analysis, we collected the Oxford Hip Score (OHS) and Hip Disability and Osteoarthritis Outcome Score (HOOS), which are patient-reported outcome measures (PROMs) for hip osteoarthritis, using the validated Japanese versions.

Reviewer #2 (3):

Results

Patient flowchart is not clear. How many patients were excluded and why?

Response:

We thank Reviewer #2 for the valuable comment.

We have clarified the patient flowchart in the manuscript, including the number of excluded patients and the reasons for exclusion.

Text Changes (Results, Lines 180-185):

Overall, 233 patients were assessed for eligibility. Among them, 60 did not have smartphones or were unable to install the application, leaving 173 patients (74.2%) with compatible smartphones. Of these, 54 declined to participate, and 119 wished to participate in the study. Among the 119 patients, data from 34 were excluded due to incomplete or invalid step-count data. Finally, data from 85 patients who underwent THA and had valid step-count data were included in the analysis (Fig 2).

Reviewer #2 (4):

Discussion

Could you elaborate more on why patients living in Solitude had more step count than cohabiting patients?

Response:

We thank the Reviewer #2 for the valuable comment. We have revised the text.

Text Changes (Discussion, Lines 274-278):

Our study found that solitary patients exhibited significantly higher step counts than cohabiting patients during the plateau phase of postoperative recovery, which spanned from postoperative week 20 to week 50, suggesting that living alone may require greater physical autonomy in daily life. This finding may provide helpful information for surgical decision-making.

Reviewer #2 (5):

Was there any relation between PROMs and step count?

Response:

We thank the Reviewer #2 for the valuable comment. We have revised the text.

Text Changes (Discussion, Lines 265-267):

Solitude patients maintained higher activity levels from the preoperative to the postoperative period than cohabiting patients did, a difference that was not captured by OHS or HOOS.

---

## [Decision Letter · Decision Letter 1]

Using Smartphone Step Counts to Monitor Patients with Total Hip Arthroplasty: The Impact of Patients’ Living Arrangements and Residential Location

PONE-D-25-08198R1

Dear Dr. Hamai,

We’re pleased to inform you that your manuscript has been judged scientifically suitable for publication and will be formally accepted for publication once it meets all outstanding technical requirements.

Kind regards,

Osama Farouk

Academic Editor

PLOS ONE

Additional Editor Comments (optional):

Reviewers' comments:

Reviewer's Responses to Questions

**Comments to the Author**

Reviewer #2: All comments have been addressed

2. Is the manuscript technically sound, and do the data support the conclusions?

Reviewer #2: Yes

3. Has the statistical analysis been performed appropriately and rigorously?

Reviewer #2: Yes

4. Have the authors made all data underlying the findings in their manuscript fully available?

Reviewer #2: Yes

5. Is the manuscript presented in an intelligible fashion and written in standard English?

Reviewer #2: Yes

Reviewer #2: (No Response)

**Do you want your identity to be public for this peer review?** For information about this choice, including consent withdrawal, please see our Privacy Policy

Reviewer #2: No

---

## [Editor Report · Acceptance letter]

PONE-D-25-08198R1

PLOS ONE

Dear Dr. Hamai,

I'm pleased to inform you that your manuscript has been deemed suitable for publication in PLOS ONE. Congratulations! Your manuscript is now being handed over to our production team.

Kind regards,

on behalf of

Dr. Osama Farouk

Academic Editor

PLOS ONE